# Identification of potential human pancreatic α-amylase inhibitors from natural products by molecular docking, MM/GBSA calculations, MD simulations, and ADMET analysis

**Santosh Basnet**[1], **Madhav Prasad Ghimire**[2☯], **Tika Ram Lamichhane**[2☯], **Rajendra Adhikari**[3☯], **Achyut Adhikari**[1☯]*

1 Central Department of Chemistry, Tribhuvan University, Kirtipur, Kathmandu, Nepal, 2 Central Department of Physics, Tribhuvan University, Kirtipur, Kathmandu, Nepal, 3 Department of Physics, Kathmandu University, Dhulikhel, Nepal

☯ These authors contributed equally to this work.
* achyutraj05@gmail.com

**Data Availability Statement:** All relevant data are within the paper and its Supporting information files.

## Abstract

Human pancreatic α-amylase (HPA), which works as a catalyst for carbohydrate hydrolysis, is one of the viable targets to control type 2 diabetes. The inhibition of α-amylase lowers blood glucose levels and helps to alleviate hyperglycemia complications. Herein, we systematically screened the potential HPA inhibitors from a library of natural products by molecular modeling. The modeling encompasses molecular docking, MM/GBSA binding energy calculations, MD simulations, and ADMET analysis. This research identified newboulaside B, newboulaside A, quercetin-3-O-β-glucoside, and sasastilboside A as the top four potential HPA inhibitors from the library of natural products, whose Glide docking scores and MM/GBSA binding energies range from -9.191 to -11.366 kcal/mol and -19.38 to -77.95 kcal/mol, respectively. Based on the simulation, among them, newboulaside B was found as the best HPA inhibitor. Throughout the simulation, with the deviation of 3Å (acarbose = 3Å), it interacted with ASP356, ASP300, ASP197, THR163, ARG161, ASP147, ALA106, and GLN63 via hydrogen bonding. Additionally, the comprehensive ADMET analysis revealed that it has good pharmacokinetic properties having not acutely toxic, moderately bioavailable, and non-inhibitor nature toward cytochrome P450. All the results suggest that newboulaside B might be a promising candidate for drug discovery against type 2 diabetes.

## 1. Introduction

Type 2 diabetes, associated with hyperglycemia, is a long-term endocrine metabolic disorder. It is most prevalent and caused by aberrant insulin activity. It causes various chronic complications. Despite medical improvements, it is still one of the leading causes of death in adults [1, 2].

**Funding:** This work was supported by the University Grants Commission, Bhaktapur, Nepal, as part of a thesis grant fellowship award (MRS-77/78-S&T-36). The funder detail is in the following links (http://www.ugcnepal.edu.np/uploads/notice/csmNgT.pdf). When the proposal was submitted to the UGC Nepal, it was proposed under the title of "In-Silico Evaluation of Different Natural Products as Inhibitors of α-Amylase Enzyme". UGC provided funding (funding number: MRS-77/78-S&T-36) as dissertation support, but the title has been modified to "Identification of potential human pancreatic α-amylase inhibitors from natural products by molecular docking, MM/GBSA calculations, MD simulations, and ADMET analysis" during journal submission to give more emphasis on finding.

**Competing interests:** The authors have declared that no competing interests exist.

Delaying the digestion of carbohydrates ameliorates hyperglycemia impacts. Multiple enzymes, including human pancreatic $\alpha$-amylase (HPA) and $\alpha$-glucosidase, are involved in the hydrolysis of carbohydrates. According to the study, in the HPA, catalytic residues (ASP197, GLU233, and ASP300) are found in an eight-stranded parallel beta-barrel in domain A (1–99, 169–404). In carbohydrate hydrolysis, ASP197 acts as a nucleophile, and GLU233 and ASP300 may act as acid/base catalysts. Therefore, one of the proven scientific methods to manage type 2 diabetes is the inhibition of HPA by ligand, which lowers postprandial blood glucose levels [3–5].

A wide variety of oral anti-diabetic medications, including acarbose and miglitol, were used by doctors to treat type 2 diabetes. The undesirable, serious side effects such as gas, bloating, and diarrhea, however, are becoming apparent. Therefore, the need for efficient, affordable, and low-toxic anti-diabetic medications is urgent [6–10].

To treat diabetes, ancient people relied on medications from natural sources [11]. Therefore, natural products derived from these sources may be an appealing strategy for controlling type 2 diabetes [12]. The research is ramping up to explore a potent HPA inhibitor. The *in vitro* study shows that different natural products exhibit α-amylase inhibition activity. There have been several *in vitro* studies on natural products up to this moment, but an efficient HPA inhibitor is still lacking. Therefore, to identify the potential HPA inhibitor from natural products, structure-based molecular modeling was employed. The modeling comprises molecular docking, MM/GBSA binding energy calculations, molecular dynamics simulations, and absorption, distribution, metabolism, excretion, and toxicity (ADMET) analysis.

This computational study proposed newboulaside B as the potential HPA inhibitor from natural products. The study also suggests a possible binding mode of the inhibitor with HPA.

## 2. Material and methods

### 2.1. Curation and preparation of ligands

Based on *in vitro* data on $\alpha$-amylase, an in-house library of 39 ligands (including acarbose as the standard ligand) was created to screen a potential HPA inhibitor. Supplementary data contains information about the ligands (S1 Table). The ligand structures were sketched in Maestro's 2D sketcher and converted to a low-energy 3D state using Schrödinger's LigPrep module [13]. The OPLS4 force field was employed to generate a low-energy state structure [14]. Additionally, at pH 7.0 ± 2.0, Epik produced tautomeric and ionization states and generated a maximum of 32 conformers per ligand.

### 2.2. Protein preparation and target validation

The structure of HPA was obtained from the RCSB protein data repository and prepared using Schrödinger's Protein Preparation Wizard module (S2 Table) [15, 16]. It was preprocessed with Prime, which assigned bond ordering, added hydrogen atoms, and filled missing loops and side chains; Epik was then run at pH 7.0 ± 2.0 [17]. Additionally, after removing hetero groups, it was optimized by optimizing hydrogen bonding and minimized by employing the OPLS4 force field [18]. The prepared protein quality and its ligand binding region were assessed and validated by the SiteMap application [19]. By incorporating catalytic residues (ASP197, GLU233, and ASP300) receptor grid was created. Further, the cognate ligand was re-docked to validate the target.

## 2.3. Molecular docking and MM/GBSA binding energy calculation

Glide, in extra precision mode (XP), was used in the molecular docking of natural products with HPA [20]. Schrödinger's prime module was employed to calculate the MM/GBSA binding energy of protein and ligand complexes. The implicit VSGB solvation model [21] and OPLS4 force field were employed to calculate the binding energy.

## 2.4. Molecular dynamics simulation

A common technique in drug discovery is rigid receptor-based molecular docking. However, pitfalls exist in it. Therefore, to study the molecular dynamics behavior of the protein-ligand complex, the 200 ns molecular dynamics simulation was performed using the Academic Desmond 2019 GPU version [22]. The molecular system was solvated using the explicit solvation model (SPC) under orthorhombic periodic boundary conditions with a buffer region of 10 Å. The default OPLS_2005 force field was used [23]. $Na^+$ and $Cl^-$ ions were used in the system builder to neutralize the system. The system was made more relaxed by doing a short-duration simulation and minimization. The Noose-Hoover thermostat method and Martyna-Tobias-Klien barostat method were employed in the NPT ensemble to maintain the temperature and pressure at 300 K and 1.01325 bar, respectively. A multiple-time step RESPA integrator was used in the dynamics study of bonded, near, and far bonded interactions.

## 2.5. Post MM/GBSA binding energy calculation

Schrödinger's module was employed to calculate the post-MM/GBSA binding energy of the top four protein and ligand complexes, where the implicit solvation model and OPLS4 force field were employed. The MM/GBSA is a vital approach for determining the binding energy of the complex because of its balance in speed and accuracy. The energy for binding of the ligand (L) to the receptor (R) to form the complex (RL) is $\Delta G_{bind} = G_R - G_L - G_{RL}$ [24–26].

## 2.6. ADMET analysis

QikProp of Schrödinger was used to calculate the pharmacokinetic characteristics of the best ligand [27]. Additionally, using admetSAR, ProTox-II, pkCSM, and the SwissADME web server, the hit's ADMET properties were analyzed [28–31].

# 3. Results and discussion

## 3.1. Protein preparation and target validation

Before performing molecular docking, human pancreatic *α*-amylase (HPA) was prepared and validated. The catalytic region's validity was confirmed using SiteMap. It predicted SiteScore, Dscore, volume, and balance of the most likely ligand-binding (catalytic) area to be 0.985, 1.010, 341.285, and 0.751, respectively, which meet the almost threshold. TRP58, TRP59, TYR62, GLN63, HIE101, TYR151, LEU162, LEU165, ARG195, ASP197, ALA198, LYS200, HIS201, GLU233, VAL234, ILE235, HIE299, ASP300, and HIE305 residues were found in this area, confirming the catalytic region as the ligand-binding region (S3 Table, S1 Fig). Further, the target, RMSD of less than 2 Å, was validated by re-docking the cognate ligand (S1 Fig). As a result, the SiteMap calculation, and cognate ligand re-docking all support the target protein's feasibility for further modeling.

## 3.2. Molecular docking and MM/GMBSA binding energy calculation

In this study, 38 molecules and acarbose were docked with HPA using Glide XP (extra precision mode). Then, MM/GBSA was employed to predict the binding energy between the protein and ligand complex. Based on the docking pose and score, the molecular docking showed that newboulaside B, newboulaside A, quercetin-3-O-β-glucoside, and sasastilboside A are the best molecules (Table 1). Among them, according to the literature, newboulaside B, newboulaside A, and sasastilboside A are all new molecules [32, 33]. The MM/GBSA approach, among them, showed that newboulaside B is the best binding molecule.

The newboulaside B docking score and MM/GBSA binding energy with HPA were predicted to be -11.366 kcal/mol (acarbose = -12.581 kcal/mol) and -50.29 kcal/mol (acarbose = -59.77 kcal/mol), respectively. It interacted with active site residues (ASP197 and ASP300) as well as additional binding site residues (ASN105, ALA106, ASP147, THR163, and ARG195) via hydrogen bonding, according to the molecular docking (Fig 1a and 1b).

The binding of HPA by newboulaside B was aided more by hydrogen bonds (-6.93) and hydrophobic enclosure reward (-0.25) than in the binding of HPA by acarbose (Fig 1c and 1d). In comparison to other parameters, in the binding, the hydrogen bond contribution was found significant. So, we speculate that inhibition of HPA by newboulaside B might be due to the formation of hydrogen bonding with active site residues ASP197 and ASP300.

When the ligand changes, it undergoes structural remodeling, resulting in a change in binding affinity. The docking score and MM/GBSA binding energy of the HPA newboulaside A complex were predicted to be -10.533 kcal/mol and -19.38 kcal/mol, respectively. ASN53, TRP59, GLN63, ASP197, GLU233, and HIE305 interacted with newboulaside A via hydrogen bonding (Fig 2a and 2b). TRP59, THR163, GLU233, and HIE305 formed hydrogen bonds with quercetin-3-O-β-glucoside, while TRP59 interacted via pi-pi interaction, with a docking score of -10.038 kcal/mol, and the MM/GBSA binding energy of -19.65 kcal/mol, respectively (Fig 2c and 2d). Similarly, THR163, ARG195, GLU233, HIE299, and ASP300 interacted with sasastilboside A via hydrogen bond, with a docking score of -9.191 kcal/mol and the MM/GBSA binding energy of -36.58 kcal/mol, respectively (Fig 2e and 2f).

## 3.3. Molecular dynamics simulation

Molecular docking predicts the binding mode between a protein and a ligand. However, it is riddled with flaws and unable to foretell the stability and dynamic behavior of the molecule during the binding process. Understanding these by experimental means is also tedious. However, by simulating molecular activity in a computational setting, we can forecast experimentally challenging molecular behavior. Therefore, the top four HPA ligand (acarbose, newboulaside B, newboulaside A, quercetin-3-O-β-glucoside, and sasastilboside A) complexes from molecular docking were subjected to the 200 ns of 1000 frame molecular dynamics simulation to confirm the hit.

Monitoring of protein secondary structure elements (SSE) revealed that the majority of strands and helices contributed 100% throughout the simulation and accounted for 33.84 percent of the total composition; the time versus residue index plot also revealed that strands near the binding pocket (ASP197, GLU233, and ASP300) contributed almost time, indicating the structure of HPA's stability. However, due to the loop, there may be a little more fluctuation around residues 130 and 370 (S2 Fig).

The molecular dynamics simulation anticipated that newboulaside B would be remarkably stable with HPA. The backbone of HPA in the newboulaside B complex was deviated by less than 2 Å. After a sharp initial deviation to 1.75 Å, it nearly stabilized. Throughout the

**Table 1. Glide molecular docking and MM/GBSA binding energy of ligands.**

| SN | Compound | docking score (kcal/mol) | glide gscore (kcal/mol) | MM/GBSA dG Bind (kcal/mol) |
|---|---|---|---|---|
| 1 | acarbose | -12.581 | -12.832 | -59.77 |
| 2 | newboulaside B | -11.366 | -11.514 | -50.29 |
| 3 | newboulaside A | -10.533 | -10.68 | -19.38 |
| 4 | quercetin-3-O-*β*-glucoside | -10.038 | -10.067 | -19.65 |
| 5 | sasastilboside A | -9.191 | -9.192 | -36.58 |
| 6 | ligularoside A | -8.910 | -8.91 | -23.11 |
| 7 | mangiferin | -8.895 | -9.274 | -33.56 |
| 8 | daucosterol | -8.411 | -8.411 | -22.33 |
| 9 | kaempferol-3-O-*β*-glucoside | -8.031 | -8.06 | -28.07 |
| 10 | rhododendron | -7.436 | -7.436 | -26.13 |
| 11 | valoneic acid dilactone | -7.48 | -7.687 | -39.38 |
| 12 | DDBT | -7.174 | -7.275 | -20.65 |
| 13 | myricetin | -7.095 | -7.132 | -34.03 |
| 14 | peruviose E | -6.894 | -6.894 | -23.57 |
| 15 | peruviose B | -6.888 | -6.888 | -26.28 |
| 16 | peruviose D | -6.881 | -6.881 | -44.62 |
| 17 | chrysoeriol | -6.655 | -6.695 | -10.06 |
| 18 | *α*-viniferin | -6.464 | -6.464 | -44.5 |
| 19 | 1-acetoxypinoresinol | -5.451 | -5.451 | -29.28 |
| 20 | emodin | -5.23 | -5.323 | -17.72 |
| 21 | momilactone B | -5.129 | -5.129 | -6.6 |
| 22 | *β*-sitosterol | -4.972 | -4.972 | -28.93 |
| 23 | 3´-geranylchalconaringenin | -4.863 | -5.362 | -12.56 |
| 24 | diosgenin | -4.786 | -4.786 | -33.8 |
| 25 | momilactone A | -4.772 | -4.772 | -29.36 |
| 26 | lupenone | -4.658 | -4.658 | -33.24 |
| 27 | rostratazine B | -4.582 | -4.582 | -19.18 |
| 28 | ursolic acid | -4.42 | -4.426 | -22.29 |
| 29 | norchelerythrine | -4.414 | -4.415 | -36.82 |
| 30 | cycloeucalenone | -4.314 | -4.314 | -6.33 |
| 31 | skimmianine | -4.151 | -4.164 | -39.66 |
| 32 | sesamine | -4.147 | -4.147 | -31.28 |
| 33 | gedunin | -4.057 | -4.057 | -17.86 |
| 34 | chaylbemide B | -4.038 | -4.038 | -8.13 |
| 35 | oleanolic acid | -3.866 | -3.872 | -5.88 |
| 36 | dehydrodieugenol B | -3.623 | -3.633 | -40.32 |
| 37 | trans-fagaramide | -3.616 | -3.616 | -5.16 |
| 38 | chaylbemide C | -3.452 | -3.452 | -11.23 |
| 39 | chaylbemide A | -2.716 | -2.716 | -16.86 |

Key: DDBT -2-(4-(3,5-dihydroxyphenoxy)-3,5-dihydroxyphenoxy) benzene-1,3,5-triol

simulation, newboulaside B deviated around 1.5 Å (acarbose = 1.5 Å) in its alignment with itself and approximately 3 Å (acarbose = 3 Å) in its alignment with the protein (Fig 3a).

In the interaction region, the C-alpha residues of the HPA newboulaside B complex fluctuated about 1 Å, but in the HPA acarbose complex, after residue 300, they fluctuated largely (Fig 3c and 3d). Throughout the simulation, newboulaside B fluctuated less than 1 Å

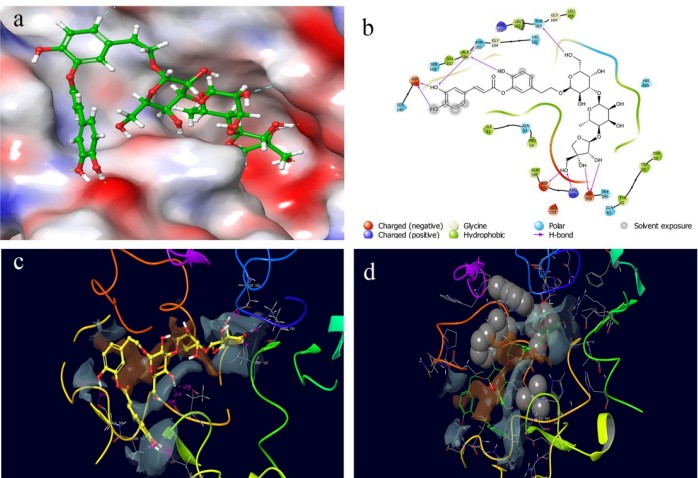

**Fig 1.** Docked poses and molecular interactions of newboulaside B in the binding site of HPA: (a) 3D molecular surface diagrams, and (b) 2D ligand interactions of newboulaside B with HPA. Glide XP molecular docking contributing term in the HPA and newboulaside B complex: (c) HBOND (hydrogen bond) between HPA and newboulaside B (pink interaction) and (d) PhobEn (hydrophobic enclosure reward) of HPA (CPK gray), and PhobEn of newboulaside B (green ball and stick). The hydrophobic/hydrophilic map is represented as gray/brown patches.

(acarbose = 1.2 Å) in its alignment with itself and less than 1.8 Å (acarbose = 2 Å) in its alignment with protein (S3 Fig). The stability of other compounds, such as newboulaside A, quercetin-3-O-β-glucoside, and sasastilboside A, was predicted to be much less than that of acarbose and newboulaside B. The observed RMSD of their fitting with protein was found more than 3.5 Å (Fig 3b). As a result, among top compounds, acarbose, and newboulaside B's stability with HPA may be comparable, as indicated by the aforementioned comparative deviations and fluctuations.

A minimum of 10 protein-ligand interactions, of which at least 5 were significant, were observed between HPA and acarbose as well as HPA and newboulaside B throughout the simulation (Fig 4a and 4c). Acarbose interacted with ASP300, ASP197, and THR163 via hydrogen bonds over 70% of the simulated duration (Fig 4b). Whereas, newboulaside B interacted with residues (ASP197, ASP147, and ALA106) over 90% of the simulation period via hydrogen

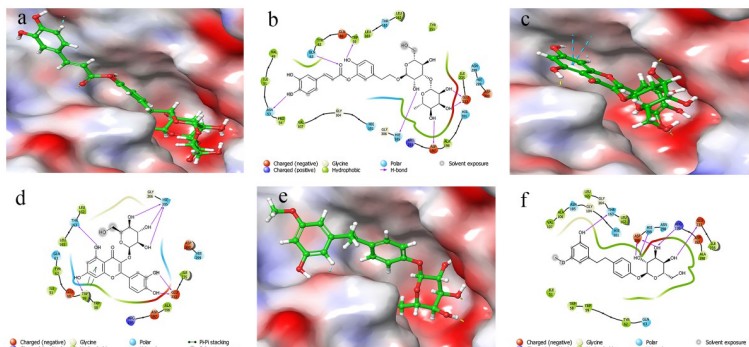

**Fig 2.** Docked poses and molecular interactions of HPA with inhibitors: 3D molecular surface diagrams and 2D ligand interactions of newboulaside A (a) and (b), quercetin-3-O-β-glucoside (c) and (d), and sasastilboside A (e) and (f), respectively. The hydrogen bonds are represented as pink lines.

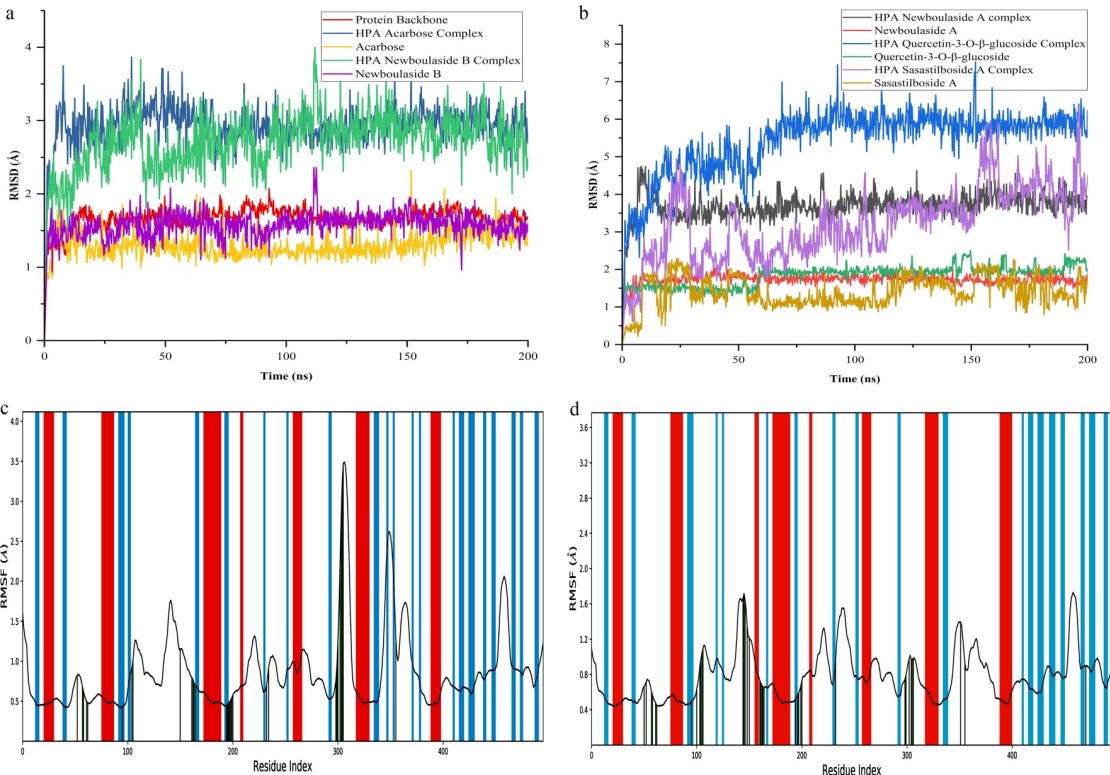

**Fig 3.** RMSD and RMSF analysis of MD simulation trajectory: The RMSD plot obtained for HPA (a) acarbose and newboulaside B complex, and (b) newboulaside A, quercetin-3-O-β-glucoside, and sasastilboside A complex. The RMSF plot was obtained for HPA (c) acarbose complex, and (d) newboulaside B complex. The strand, helix, and loop region of HPA are represented as blue, red, and white, respectively.

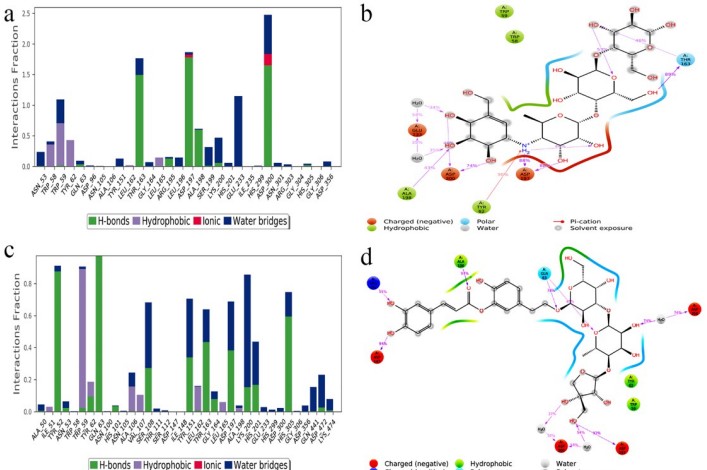

**Fig 4.** Analysis of inhibitor's molecular interaction and type of contact with HPA after MD simulations: Normalized stacked bar chart of HPA binding site residues interacting with (a) acarbose and (c) newboulaside B via hydrogen bonds, hydrophobic and ionic interactions, and water bridges. Detailed schematic interaction of (b) acarbose and (d) newboulaside B atoms with the binding site residues of HPA. Interactions happening more than 30% of the simulation times are shown.

bonding. Further, catalytic residue (ASP300) also interacted via multiple hydrogen bonding (Fig 4d).

The timeline charts also showed that at least three bonds in the HPA complex of acarbose and newboulaside B are almost unbroken and interacted significantly throughout the simulation. THR163, ASP197, and ASP300 interacted via more or less uninterrupted bonding with acarbose, while TRP59 and GLU233 interacted via partial bonding (S4 Fig). Similarly, newboulaside B interacted via nearly continuous bonding with ALA106, ASP147, and ASP197, and partial bonding with GLN63, ARG161, THR163, ASP300, and ASP356 (S4 Fig). These interactions might help to prevent them from diffusing out from the ligand-binding pocket. This evidence substantiates the hypothesis that newboulaside B inhibits HPA.

However, newboulaside A, quercetin-3-O-*β*-glucoside, and sasastilboside A demonstrated fewer interactions with active site residues than newboulaside B, according to the protein-ligand interaction analysis (S5 Fig). Further, the HPA complexes of these three were more unstable than the complex of newboulaside B, according to the ligand-protein interaction diagram of MD simulation (S5 Fig). Overall, they exhibited significantly poorer interactions with HPA than acarbose and newboulaside B, as evidenced by the high deviations and weak interactions.

The outcome of the HPA newboulaside B complex's molecular docking and MM/GBSA binding energy is supported by the molecular dynamics simulation, too. Thus, based on comparative stability, binding affinity, and interactions with acarbose, newboulaside B is expected to be effective in inhibiting HPA.

Existing knowledge also reinforces our computational analysis. *Newbouldia laevis* (P. Beauv.), which is used in African traditional medicine to treat a variety of illnesses such as sickle cell, diarrhea, inflammation, aches, tumors, skin problems, hypertension, and diabetes, is reported to be the natural source of newboulaside B [34]. Ethanolic extracts and aqueous root extracts of the plant exhibited anti-diabetic activity in streptozocin-induced diabetic Wistar rats and alloxan-induced diabetic rats, respectively [35, 36]. Ethanolic extracts and aqueous leaf extracts of the plant reported antioxidant activity in diabetic rats and insulin-inducing activity [37, 38]. The methanolic extract of the plant suppressed induced diabetes by 60.2 percent [39].

Two new caffeic acid glycosides (newboulaside A and newboulaside B) isolated from ethanolic extracts of the *Newbouldia laevis*'s leaves inhibited *α*-amylase with $IC_{50}$ values of 4.95 and 4.44 μg/mL, respectively (acarbose $IC_{50}$ = 4.05 μg/mL) [32]. The experiment was done on fungal α-amylase. Therefore to correlate stability with human pancreatic α-amylase (2QV4), we also carried out the structural similarity and molecular dynamics simulation on fungal α-amylase (7TAA) [40, 41]. The structural similarity, with the exception in the loop region, indicates that the active residues of 2QV4 (ASP197, GLU233, ASP300) and the active residues of 7TAA (ASP206, GLU230, ASP297) are almost aligned (S6 Fig). We also observed from the simulation that fungal α-amylase is stable with newboulaside B, but to a lesser extent than HPA, demonstrating that newboulaside B is also stable with HPA (S7 Fig).

## 3.4. Post MM/GMBSA binding energy calculation

The top four protein and ligand complexes' post-MM/GBSA binding energies at frames 1, 500, and 1000 of the 200 ns simulation were computed after the molecular dynamics simulation. The best molecule was newboulaside B, as evidenced by the post-binding energy (S4 Table). The post-binding energy was found to be increased, which might be due to the change in orientation of the complex during the simulation.

### 3.5. ADMET analysis

The drug should be highly selective and have as few side effects as possible. Due to effectiveness and unavoidable side effects, the therapeutic candidate has a high attrition rate at the late stage. ADMET problems arise due to the complexity of the biological system. The *in silico* study can help to reduce the risk of a drug candidate failure. As a result, as post-processing, identified hit (newboulaside B) drug-likeness and ADMET properties were analyzed comprehensively.

The ethanolic extract of *Newbouldia laevis* was proven to be safe in terms of acute toxicity [42]. The assessment of the ethanolic extract of the plant revealed a low toxicity profile [43]. This finding supports the traditional uses of the plant against diabetes. However, research on the pure molecule was lacking. As a result, several models, including QikProp, admetSAR, Pro-Tox-II, pkCSM, and SwissADME, were utilized to analyze the hit's pharmacokinetic properties (S5–S9 Tables).

The QikProp computed lipophilicity (QPlogPo/w) of the hit -2.995 (-2.0 to 6.5), which indicates that the hit is more hydrophilic; this low lipophilicity may decrease absorption and distribution due to the lipoid nature of the cell membrane. Similarly, it predicted that hit is aqueous soluble with a solubility value of -2.52 (-6.5 to 0.5) [44, 45]. The water solubility of the hit is due to the hydroxyl group, which contributes to absorption. The hit is comparable to acarbose, according to molecular modeling; however, it defies recognized drug-likeness, it is partly due to its large molecular size.

The SwissADME projected that the hit would have poor intestinal absorption. Similarly, it predicted the hit as less skin permeable with LogKp value of -12.08 cm/s (acarbose = -16.30 cm/s). The pkCSM predicted CNS permeability (logPS) -5.747 (logPS<-3), which indicates that the hit is impermeable to CNS. The strong polarity, high molecular weight, and poor lipophilicity of this compound may be the cause of its limited absorption. QikProp's low human serum albumin binding value of the hit (-2.173) indicates the unbound molecule can easily reach the target [46]. As a result of the large molecular size, the hit may be moderately bioavailable on average.

The ProTox-II model projected the hit toxicity would be class V (slightly toxic), with an $LD_{50}$ value of 5000 mg/kg, and also predicted that it is inactive for carcinogenicity, mutagenicity, and cytotoxicity. Additionally, it was evaluated with negative mitochondrial, carcinogenicity, and nephrotoxicity by the admetSAR, which indicates it is not acutely toxic.

The hit was predicted as negative skin sensitization by the admetSAR, which reveals that the molecule would not show a cutaneous inflammatory reaction. It also predicted that the hit would not inhibit CYP 2D6, CYP 1A2, CYP 2C19, CYP 2C9, and CYP 3A4. This property helps to metabolize xenobiotics and reduce the adverse consequences of drug-drug interactions. Similarly, the admetSAR projected that the hit would be the non-inhibitor of OCT2, OATP2B1, and MATE1. Non-inhibition of OCT2, OATP2B1, and MATE1 helps to distribute and excrete the second drug and reduces drug-drug interaction detrimental. Overall ADMET profile of newboulaside B might be good.

## 4. Conclusions

The results of the current study imply that the computational approach effectively explores the potent human pancreatic *α*-amylase (HPA) inhibitor. Herein, the computational workflow, with almost similar binding affinity and stability to acarbose, identifies newboulaside B as the potential molecule for HPA inhibition. The study also shows a good ADMET profile for the molecule and reinforces prior reports. The data presented here can be helpful for drug discovery against type 2 diabetes.

## Supporting information

**S1 Fig. Pictorial representation of HPA with cognate ligand (green) and re-docked cognate ligand (white).** The yellow and blue patches represent the SiteMap application view.
(TIF)

**S2 Fig. Analysis of secondary structure element (SSE) of HPA after MD simulation: The strand, helix, and loop region of HPA are represented as blue, red, and white, respectively.**
(TIF)

**S3 Fig. RMSF of inhibitor's atoms after MD simulation: (a) acarbose and (b) newboulaside B.**
(TIF)

**S4 Fig. Timeline representation of protein-ligand contact of MD simulation trajectory: (a) HPA acarbose complex (b) HPA newboulaside B complex.**
(TIF)

**S5 Fig.** Analysis of inhibitor's molecular interaction and type of contact with HPA after MD simulations: Normalized stacked bar chart of HPA binding site residues interacting with (a) newboulaside A, (c) quercetin-3-O-*β*-glucoside, and (e) sasastilboside A via hydrogen bonds, hydrophobic and ionic interactions and water bridges. Detailed schematic interaction of (b) newboulaside A, (d) quercetin-3-O-*β*-glucoside, and (f) sasastilboside A atoms with the binding site residues of HPA. Interactions happening more than 30% of the simulation times are shown.
(TIF)

**S6 Fig. The structural similarity in the active site of fungal α-amylase (7TAA) and human pancreatic α-amylase (2QV4).**
(TIF)

**S7 Fig. RMSD analysis of MD simulation trajectory: The RMSD plot obtained for fungal α-amylase (7TAA) with acarbose and newboulaside B.**
(TIF)

**S1 Table. Ligand library of natural products.**
(DOCX)

**S2 Table. Human pancreatic *α*-amylase.**
(DOCX)

**S3 Table. SiteMap result.**
(DOCX)

**S4 Table. Post MM/GBSA (kcal/mol) binding energy of ligands.**
(DOCX)

**S5 Table. ADMET properties of the acarbose and newboulaside B by QikProp.**
(DOCX)

**S6 Table. ADMET properties of newboulaside B and acarbose by admetSAR.**
(DOCX)

**S7 Table. Toxicity profile of newboulaside B and acarbose by ProTox-II.**
(DOCX)

**S8 Table. ADMET properties of newboulaside B (1) and acarbose (2) by pkCSM.**
(DOCX)

**S9 Table. ADMET properties of newboulaside B and acarbose by SwissADME.**
(DOCX)

## Acknowledgments

The author acknowledges Schrödinger Inc. for the software license support for the thesis research work. Similarly, the author acknowledges the Central Department of Physics, Tribhuvan University for providing the supercomputing facility supported by the Alexander von Humboldt Foundation, and IFW-Dresden, Germany. S. Basnet expresses gratitude to Schrödinger Inc. for providing training and fellowship in computational antibody engineering, rational drug design, and high-throughput virtual screening. R. Adhikari acknowledges NVIDIA for GPU grants. A. Adhikari gratefully acknowledges the Higher Education Reform Project (HERP) of Tribhuvan University for the startup grant.

## Author Contributions

**Conceptualization:** Santosh Basnet, Achyut Adhikari.

**Data curation:** Santosh Basnet.

**Formal analysis:** Santosh Basnet, Madhav Prasad Ghimire, Tika Ram Lamichhane, Rajendra Adhikari, Achyut Adhikari.

**Funding acquisition:** Santosh Basnet.

**Investigation:** Santosh Basnet.

**Methodology:** Santosh Basnet.

**Project administration:** Achyut Adhikari.

**Resources:** Santosh Basnet, Madhav Prasad Ghimire, Rajendra Adhikari, Achyut Adhikari.

**Software:** Santosh Basnet, Achyut Adhikari.

**Supervision:** Achyut Adhikari.

**Validation:** Santosh Basnet.

**Visualization:** Santosh Basnet, Tika Ram Lamichhane.

**Writing – original draft:** Santosh Basnet.

**Writing – review & editing:** Madhav Prasad Ghimire, Tika Ram Lamichhane, Rajendra Adhikari, Achyut Adhikari.

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
