## [Decision Letter · Decision Letter 0]

25 Oct 2022

PONE-D-22-26420Identification of potential human pancreatic α-amylase inhibitors from natural products by molecular docking, MM/GBSA calculations, MD simulations, and ADMET analysisPLOS ONE

Dear Dr. Basnet,

Thank you for submitting your manuscript to PLOS ONE. After careful consideration, we feel that it has merit but does not fully meet PLOS ONE’s publication criteria as it currently stands. Therefore, we invite you to submit a revised version of the manuscript that addresses the points raised during the review process.

ACADEMIC EDITOR: I have carefully evaluated your manuscript and the reviewers’ reports, and the reports indicate that major revisions are necessary. Please submit a revised manuscript which addresses all of the reviewers’ comments.

Further peer review of your revised manuscript may be needed. When you submit your revised manuscript, please include a point by point response to the reviewers’ comments and highlight the changes you have made.

Full details of the files you need to submit as well as the reviewers comments are listed at the end of this email.

We look forward to receiving your revised manuscript.

Kind regards,

Taye Beyene Demissie, PhD

Academic Editor

PLOS ONE

Journal Requirements:

"Santosh

MRS-77/78-S&T-36

Santosh Basnet

https://www.ugcnepal.edu.np/

NO"

4. Thank you for stating the following in the Funding Section of your manuscript: 

"This work was supported by the University Grants Commission, Bhaktapur, Nepal, as part of a thesis grant fellowship award (MRS-77/78-S&T-36). "

"Santosh

MRS-77/78-S&T-36

Santosh Basnet

https://www.ugcnepal.edu.np/

NO"

Reviewers' comments:

Reviewer's Responses to Questions

**Comments to the Author**

1. Is the manuscript technically sound, and do the data support the conclusions?

Reviewer #1: Partly

Reviewer #2: Partly

2. Has the statistical analysis been performed appropriately and rigorously? 

Reviewer #1: N/A

Reviewer #2: Yes

3. Have the authors made all data underlying the findings in their manuscript fully available?

Reviewer #1: Yes

Reviewer #2: Yes

4. Is the manuscript presented in an intelligible fashion and written in standard English?

Reviewer #1: Yes

Reviewer #2: Yes

5. Review Comments to the Author

Reviewer #1: 1.acarbose is a very successful product in clinical application. In order to find new antidiabetic drugs based on α-amylase, it is necessary to consider the existing problems of acarbose in the design of the manuscript. The results of the present comparison with acarbose are not particularly significant.

2.acarbose directly inhibits α-glycosidase in the intestine, and this paper is to find an inhibitor of α-amylase. Although these two enzymes have similarities, it is a question to take acarbose as a positive control. It is recommended to use the compounds with the inhibitory effect of α-amylase confirmed in the literature as a positive control.

3.Without entering the blood, the drug can directly inhibit α-amylase or α-glycosidase in the intestine to play a hypoglycemic effect. The ideal result is that the drug does not be absorbed into the blood. The ADMET result in the paper is not suitable.

4.According to the MM/GBSA results, only newboulaside B had a binding free energy similar to acarbose. The other three components ( newboulaside A, quercetin-3-O-β-glucoside, and sasastilboside A) have good docking scores but not free energies. It is suggested to use MD results to calculate the binding free energy and verify the binding ability of these components with α-amylase.

Reviewer #2: Manuscript No. : PONE-D-22-26420

Identification of potential human pancreatic α-amylase inhibitors from natural products by molecular docking, MM/GBSA calculations, MD simulations, and ADMET analysis

The authors have carried out extensive in silico studies on the interactions of newboulaside B as a human pancreatic amylase (HPA) inhibitor. The computational study has been well designed and analysed. After target protein (HPA) and newboulaside B preparation and energy minimization, molecular docking was carried out and MM/GBSA binding energies calculated using GLIDE. Further MD simulations taking into consideration the solvation of Na+ and Cl- ions it was ascertained that newboulaside B interaction with HPA was fairly stable with minimal deviation and interacted with the active site residues. Further the ADMET studies have also been carried out to check for its toxicity .

However, it is to be noted that newboulaside B inhibited fungal amylase with an IC 50 of 4.44 µg/mL ( ref no. 28). No information on its inhibition or kinetics is available on human pancreatic amylase (HPA) . To validate the in silico data it is necessary to carry out the enzyme inhibition and its kinetics to prove it as a completive inhibitor which the authors are stating. It has been very often seen that the mode of inhibition (competive, uncompetietive , non-competietive) in experimental studies differs from computational data. Hence while computational studies suggest good leads and probable modes of binding, it it relevant to validate their data experimentally.

6. PLOS authors have the option to publish the peer review history of their article (what does this mean?). If published, this will include your full peer review and any attached files.

Reviewer #1: **Yes: **Weiren Xu

Reviewer #2: No

---

## [Author Response · Author response to Decision Letter 0]

14 Jan 2023

13/01/2023

To 

The Reviewers

PLOS ONE

Subject: About the Responses to the Comments of Reviewers

Dear Reviewer,

Herein, we are submitting our revised research manuscript entitled “Identification of potential human pancreatic α-amylase inhibitors from natural products by molecular docking, MM/GBSA calculations, MD simulations, and ADMET analysis” for publication in your prestigious journal, PLOS ONE. We have highlighted where we responded to the reviewer's comments in this revision.

First of all, we would like to express our gratitude for the reviewer's insightful suggestions. In this study, we have proposed Newboulaside B as the potential inhibitor of human pancreatic α-amylase. It was isolated in 2020 from Newbouldia laevis's leaves. Previously, researchers had done in-vitro on this molecule on fungal α-amylase (https://www.tandfonline.com/doi/full/10.1080/14786419.2020.1799362). However, there was a lack of research on human pancreatic α-amylase (HPA). Therefore, we short-listed this compound for our research.

Journal Requirements:

The manuscript is presented in the format specified by the journal.

This work was supported by the University Grants Commission, Bhaktapur, Nepal, as part of a thesis grant fellowship award (MRS-77/78-S&T-36).

The funder detail is in the following links. When the proposal was submitted to the UGC Nepal, it was proposed under the title of “In-Silico Evaluation of Different Natural Products as Inhibitors of Alpha-Amylase Enzyme”. UGC provided funding (funding number: MRS-77/78-S&T-36) as dissertation support, but the title has been modified to “Identification of potential human pancreatic α-amylase inhibitors from natural products by molecular docking, MM/GBSA calculations, MD simulations, and ADMET analysis” during journal submission to give more emphasis on finding.

http://www.ugcnepal.edu.np/uploads/notice/csmNgT.pdf

3. Please state what role the funders took in the study.

The funders had no role in the study design, data collection, analysis, decision to publish, or preparation of the manuscript.

4. Please remove any funding-related text from the manuscript and let us know how you would like to update your Funding Statement.

The funding information was not included in the original manuscript and is now only mentioned in the section on the funding statement.

Comments to the Author

1. Is the manuscript technically sound, and do the data support the conclusions?

Reviewer 1 & Reviewer 2: Partly

We have correlated our computational findings with earlier experimental data in the publication. In the course of the investigation, we also performed a duplicate study to replicate the data. 

2. Has the statistical analysis been performed appropriately and rigorously?

Reviewer 1:

Thank you for your kind words, reviewer 1. We have used the appropriate tools to analyze the data in accordance with our knowledge and general scientific trends.

Reviewer 2: No comment

3. Have the authors made all data underlying the findings in their manuscript fully available?

Reviewer 1: No comment

Reviewer 2: No comment

4. Is the manuscript presented in an intelligible fashion and written in standard English?

Reviewer 1: No comment

Reviewer 2: No comment

5. Review Comments to the Author

Reviewer 1:

We appreciate the reviewer 1's insightful remarks. Your concern has been addressed in the following ways:

1. Acarbose is a very successful product in clinical application. In order to find new antidiabetic drugs based on α-amylase, it is necessary to consider the existing problems of acarbose in the design of the manuscript. The results of the present comparison with acarbose are not particularly significant.

Thank you for your valuable remarks. Acarbose is one of the effective candidates to control type 2 diabetes, reviewer 1 is correct. We also utilized it as the positive control because of this (reference compound ). Despite the candidate's success, type 2 diabetes patients continue to be an issue. They are not entirely recovering. Acarbose also shows certain adverse effects in the abdomen. Therefore our hypothesis was to search for an alternative candidate. To some extent, we tried to address it in the manuscript by adding reference 10 (references 6-10). In our opinion, this literature will be sufficient, which will also help to make the manuscript concise.

2. Acarbose directly inhibits α-glycosidase in the intestine, and this paper is to find an inhibitor of α-amylase. Although these two enzymes have similarities, it is a question to take acarbose as a positive control. It is recommended to use the compounds with the inhibitory effect of α-amylase confirmed in the literature as a positive control. Acarbose as the positive control

We appreciate your kind words. Yes, acarbose is used as a positive control against α-glucosidase. According to the literature (taken co-crystalized target (2QV4) and supplementary references (Table S1 and Table S2)), it also inhibits α-amylase. We don't think it needs to be written in the manuscript because doing so would make it less concise.

3. Without entering the blood, the drug can directly inhibit α-amylase or α-glycosidase in the intestine to play a hypoglycemic effect. The ideal result is that the drug does not be absorbed into the blood. The ADMET result in the paper is not suitable.

Thank you for the insightful feedback. Yes, the inhibitor's role is to inhibit α-amylase. However, we cannot assert that the body does not absorb the substances we ingest. We cannot directly state that our body's efflux inhibitor appears in the form of faeces. Due to this, we also included ADMET analysis in our research.

4. According to the MM/GBSA results, only newboulaside B had a binding free energy similar to acarbose. The other three components ( newboulaside A, quercetin-3-O-β-glucoside, and sasastilboside A) have good docking scores but not free energies. It is suggested to use MD results to calculate the binding free energy and verify the binding ability of these components with α-amylase.

We appreciate your insightful comment regarding the post-MM/GBSA binding energy. Using implicit solvent (VSGB), we have computed binding energy at frames 1, 500, and 1000. Due to orientation change after simulation, we observed binding energies greater than pre-MM/GBSA binding energies. From this, we also found neboulaside B as potential candidate. Updated information for all frames is highlighted in the manuscript's sections 2.5 and 3.4 as well as the supplementary material (Table S4). The number of the supplementary table has also been changed as a result of the inclusion of the post-MM/GBSA data (Table S4).

Reviewer 2:

The authors have carried out extensive in silico studies on the interactions of newboulaside B as a human pancreatic amylase (HPA) inhibitor. The computational study has been well designed and analysed. After target protein (HPA) and newboulaside B preparation and energy minimization, molecular docking was carried out and MM/GBSA binding energies calculated using GLIDE. Further MD simulations taking into consideration the solvation of Na+ and Cl- ions it was ascertained that newboulaside B interaction with HPA was fairly stable with minimal deviation and interacted with the active site residues. Further the ADMET studies have also been carried out to check for its toxicity .

However, it is to be noted that newboulaside B inhibited fungal amylase with an IC 50 of 4.44 µg/mL ( ref no. 28). No information on its inhibition or kinetics is available on human pancreatic amylase (HPA) . To validate the in silico data it is necessary to carry out the enzyme inhibition and its kinetics to prove it as a completive inhibitor which the authors are stating. It has been very often seen that the mode of inhibition (competive, uncompetietive , non-competietive) in experimental studies differs from computational data. Hence while computational studies suggest good leads and probable modes of binding, it it relevant to validate their data experimentally.

For your insightful criticism of the paper, reviewer 2 is gratefully acknowledged.

Previously, newboulaside B’s experiment was done with fungal α-amylase with IC50 value (4.44 µg/mL). We encountered a challenge when trying to conduct research with these compounds and human pancreatic α-amylase. The compound's source, which was recently determined and is also stated in the document, is Africa. Geographical constraints made it difficult for us to conduct the experiment at this time. Therefore, we tried to address such concerns by correlating them with previous experiments. 

To address correlation concern, we also did a simulation study with fungal α-amylase (7TAA), and we also correlated the active site region of human pancreatic α-amylase and fungal α-amylase. Active residues are nearly aligned, as evidenced by structural similarities, with the exception of the loop. The addition of references clarifies them as well (manuscript references: 37 and 38). Newboulaside B is more stable with HPA than fungal α-amylase, according to the simulation results, which are updated in the text (section 3.4) and the supplemental material (Fig. S6 and Fig. S7) as well. These findings support the HPA-inhibiting properties of newboulaside B.

In addition to the revision described above, the abstract, conclusions, and supporting information have all undergone minor revisions, which are mostly highlighted in the text.

Based on the above-mentioned new finding, we think our work is suitable for publication in your esteemed journal. we thus request your kind consideration and necessary action.

I hereby declare that all the authors involved in this work have approval for the submission process.

Achyut Adhikari, PhD

Associate Professor, 

Central Department of Chemistry, Tribhuvan University, 

Kirtipur-44613, Kathmandu, Nepal

Email: achyutraj05@gmail.com; Mobile Number: +977-9861490245

---

## [Decision Letter · Decision Letter 1]

31 Jan 2023

PONE-D-22-26420R1Identification of potential human pancreatic α-amylase inhibitors from natural products by molecular docking, MM/GBSA calculations, MD simulations, and ADMET analysisPLOS ONE

Dear Dr. Basnet,

Thank you for submitting your manuscript to PLOS ONE. After careful consideration, we feel that it has merit but does not fully meet PLOS ONE’s publication criteria as it currently stands. Therefore, we invite you to submit a revised version of the manuscript that addresses the points raised during the review process.

The reviewers raised issues which need to be addressed. Please submit your revised manuscript by Mar 17 2023 11:59PM. If you will need more time than this to complete your revisions, please reply to this message or contact the journal office at plosone@plos.org. Please include the following items when submitting your revised manuscript:A rebuttal letter that responds to each point raised by the academic editor and reviewer(s). You should upload this letter as a separate file labeled 'Response to Reviewers'.A marked-up copy of your manuscript that highlights changes made to the original version. You should upload this as a separate file labeled 'Revised Manuscript with Track Changes'.An unmarked version of your revised paper without tracked changes. You should upload this as a separate file labeled 'Manuscript'.If applicable, we recommend that you deposit your laboratory protocols in protocols.io to enhance the reproducibility of your results. Protocols.io assigns your protocol its own identifier (DOI) so that it can be cited independently in the future. For instructions see: https://journals.plos.org/plosone/s/submission-guidelines#loc-laboratory-protocols. Additionally, PLOS ONE offers an option for publishing peer-reviewed Lab Protocol articles, which describe protocols hosted on protocols.io. Read more information on sharing protocols at https://plos.org/protocols?utm_medium=editorial-email&utm_source=authorletters&utm_campaign=protocols.

We look forward to receiving your revised manuscript.

Kind regards,

Taye Beyene Demissie, PhD

Academic Editor

PLOS ONE

Journal Requirements:

Reviewers' comments:

Reviewer's Responses to Questions

**Comments to the Author**

1. If the authors have adequately addressed your comments raised in a previous round of review and you feel that this manuscript is now acceptable for publication, you may indicate that here to bypass the “Comments to the Author” section, enter your conflict of interest statement in the “Confidential to Editor” section, and submit your "Accept" recommendation.

Reviewer #1: (No Response)

Reviewer #2: All comments have been addressed

2. Is the manuscript technically sound, and do the data support the conclusions?

Reviewer #1: Partly

Reviewer #2: Yes

3. Has the statistical analysis been performed appropriately and rigorously? 

Reviewer #1: Yes

Reviewer #2: Yes

4. Have the authors made all data underlying the findings in their manuscript fully available?

Reviewer #1: Yes

Reviewer #2: Yes

5. Is the manuscript presented in an intelligible fashion and written in standard English?

Reviewer #1: (No Response)

Reviewer #2: Yes

6. Review Comments to the Author

Reviewer #1: 1. Desmond has a FEP module, which can calculate the free energy according to the MD trajectory. Is it appropriate to use the Post MM/GBSA binding energy method ? It is suggested to provide the literature for using a similar calculation method.

2. The top structure found in this manuscript has been reported in literature experiments with similar activity . If the interaction mode between ligand and target cannot be deeply analyzed to provide direction for the discovery of new structures, the value of the manuscript will be low. Please discuss this in the manuscript together with appropriate references.

Reviewer #2: The authors have addressed the queries in the revised manuscript. The authors have identified 4 potential HPA inhibitors using an in silico approach. Further ADMET analysis on the molecules ahs also been performed. The work has been carried out thoroughly though validation of the molecules by experimental methods would have added value to the paper.

---

## [Author Response · Author response to Decision Letter 1]

19 Feb 2023

20/02/2023

To 

The Reviewer

PLOS ONE

Subject: About the Responses to the Comments of Reviewers (Minor Revision)

Dear Reviewer,

First of all, we would like to express our gratitude for the reviewer's insightful suggestions and kind words. Herein, we are submitting our revised research manuscript entitled “Identification of potential human pancreatic α-amylase inhibitors from natural products by molecular docking, MM/GBSA calculations, MD simulations, and ADMET analysis” for publication in your prestigious journal, PLOS ONE. We have responded to the reviewers' comments with track changes.

In this study, we have proposed Newboulaside B as the potential inhibitor of human pancreatic α-amylase. It was isolated in 2020 from Newbouldia laevis's leaves. Previously, researchers had done in-vitro on this molecule on fungal α-amylase (https://www.tandfonline.com/doi/full/10.1080/14786419.2020.1799362). However, there was a lack of research on human pancreatic α-amylase (HPA). Therefore, we also short-listed this compound for our research. Our study also correlates the candidate with the previous experiment. Therefore, in the future, we hope that this molecule will be useful in the pharmaceutical industry as a lead/optimized molecule for drug discovery.

Comments to the Author

1. If the authors have adequately addressed your comments raised in a previous round of review and you feel that this manuscript is now acceptable for publication, you may indicate that here to bypass the “Comments to the Author” section, enter your conflict of interest statement in the “Confidential to Editor” section, and submit your "Accept" recommendation.

Reviewer #1: (No Response)

Reviewer #2: All comments have been addressed

2. Is the manuscript technically sound, and do the data support the conclusions?

Reviewer #1: Partly

Reviewer #2: Yes

3. Has the statistical analysis been performed appropriately and rigorously?

Reviewer #1: Yes

Reviewer #2: Yes

4. Have the authors made all data underlying the findings in their manuscript fully available?

Reviewer #1: Yes

Reviewer #2: Yes

5. Is the manuscript presented in an intelligible fashion and written in standard English?

Reviewer #1: (No Response)

Reviewer #2: Yes

6. Review Comments to the Author

Reviewer 1:

We appreciate reviewer 1's insightful remarks. Your concern has been addressed in the following ways:

1. Desmond has a FEP module, which can calculate the free energy according to the MD trajectory. Is it appropriate to use the Post MM/GBSA binding energy method? It is suggested to provide the literature for using a similar calculation method.

Thank you for your valuable remarks regarding the FEP module of Desmond for the post-analysis. Yes, we agree with your comment’s intention. We must say that for the post-analysis, despite being expensive, the FEP of Schrodinger is the better technique than MM/GBSA approach. Unfortunately, we must say that recently our license date of Schrodinger has expired, so we became unable to calculate FEP from Desmond. Therefore, we have some compulsion to rely on the previously followed approach [i.e. MM/GBSA approach]. Yes, MM/GBSA is also an appropriate tool for the post-binding energy analysis, which has been clarified in the manuscript by adding references [24,25,26] in section 2.5 (post-MM/GBSA binding energy calculation).

2. The top structure found in this manuscript has been reported in literature experiments with similar activity. If the interaction mode between ligand and target cannot be deeply analyzed to provide direction for the discovery of new structures, the value of the manuscript will be low. Please discuss this in the manuscript together with appropriate references.

Thank you for your insightful remarks. Yes, we have analyzed our top molecule (newboulaside B) interaction deeply. In the manuscript’s molecular docking section 3.2 (paragraphs 1, 2, 3 with reference 32), we have analyzed molecular interaction. 

We all know that molecular docking is not an accurate and sufficient tool to analyze binding interaction. In the actual environment, molecules do not behave rigidly as in docking, they behave flexibly. Therefore, they may interact differently. Thus, these interactions were further analyzed deeply in the molecular dynamics section 3.3 (paragraphs 4, 5, 8) in every frame of the simulation by using the stacked bar, timeline chart, and the interaction percentage time. Therefore, the interaction of protein and ligand, by correlating, has been analyzed in both rigid and flexible forms profoundly.

Reviewer 2:

The authors have addressed the queries in the revised manuscript. The authors have identified 4 potential HPA inhibitors using an in silico approach. Further ADMET analysis on the molecules has also been performed. The work has been carried out thoroughly though validation of the molecules by experimental methods would have added value to the paper.

Based on the above-mentioned new finding, we think our work is suitable for publication in your esteemed journal. we thus request your kind consideration and necessary action.

I hereby declare that all the authors involved in this work have approval for the submission process.

Achyut Adhikari, PhD

Associate Professor, 

Central Department of Chemistry, Tribhuvan University, 

Kirtipur-44613, Kathmandu, Nepal

Email: achyutraj05@gmail.com; Mobile Number: +977-9861490245

---

## [Editor Report · Decision Letter 2]

21 Feb 2023

Identification of potential human pancreatic α-amylase inhibitors from natural products by molecular docking, MM/GBSA calculations, MD simulations, and ADMET analysis

PONE-D-22-26420R2

Dear Dr. Basnet,

We’re pleased to inform you that your manuscript has been judged scientifically suitable for publication and will be formally accepted for publication once it meets all outstanding technical requirements.

Kind regards,

Taye Beyene Demissie, PhD

Academic Editor

PLOS ONE

---

## [Editor Report · Acceptance letter]

7 Mar 2023

PONE-D-22-26420R2 

Identification of potential human pancreatic *α*-amylase inhibitors from natural products by molecular docking, MM/GBSA calculations, MD simulations, and ADMET analysis 

Dear Dr. Basnet:

I'm pleased to inform you that your manuscript has been deemed suitable for publication in PLOS ONE. Congratulations! Your manuscript is now with our production department. 

Kind regards, 

on behalf of

Dr. Taye Beyene Demissie 

Academic Editor

PLOS ONE